# Effect of Antioxidant Supplementation on Markers of Oxidative Stress and Muscle Damage after Strength Exercise: A Systematic Review

**DOI:** 10.3390/ijerph19031803

**Published:** 2022-02-05

**Authors:** Cristina Canals-Garzón, Rafael Guisado-Barrilao, Darío Martínez-García, Ignacio Jesús Chirosa-Ríos, Daniel Jerez-Mayorga, Isabel María Guisado-Requena

**Affiliations:** 1Department of Exercise Medicine, Cardiovascular Risk and Rehabilitation, Faculty of Health Sciences, University of Granada, 18016 Granada, Spain; nevada1755@hotmail.com (C.C.-G.); rguisado@ugr.es (R.G.-B.); 2Faculty of Rehabilitation Sciences, Universidad Andres Bello, Concepción 7591538, Chile; damaga1991@gmail.com (D.M.-G.); daniel.jerez@unab.cl (D.J.-M.); 3Department of Physical Education and Sports, Faculty of Sport Sciences, University of Granada, 18011 Granada, Spain; 4Departament of Nursing, Physiotherapy and Occupational Therapy, Faculty of Nursing, Group of Preventive Activities in the University Health Sciences Setting, University of Castilla-La Mancha (Universidad de Castilla-La Mancha/UCLM), 02071 Albacete, Spain; isabelm.guisado@uclm.es

**Keywords:** strength training, oxidative stress, muscle damage, free radicals, antioxidants

## Abstract

Background: The purpose of this systematic review was to determine the effect of antioxidant consumption on markers of oxidative stress and muscle damage after performing a muscle strength exercise. Methods: The Preferred Reporting Items for Systematic Review and Meta-Analyses (PRISMA) statements were followed. Four databases were used: Scopus, PubMed, WOS and SportDiscus. Methodological quality was assessed using the PEDro scale. Results: A total of 1709 articles were retrieved and following duplicate removal and application of exclusion criteria seven articles were reviewed. Supplementation with pomegranate juice alleviates oxidative stress, taurine reduces muscle damage, melatonin protects the skeletal muscles, blueberries decrease oxidation and oats mitigate muscle damage. Conclusions: Acute administration of antioxidants immediately before or during an exercise session can have beneficial effects, such as delay of fatigue and a reduction in the recovery period. Administration of antioxidant susbtances may reduce muscle damage and oxidative stress markers.

## 1. Introduction

Physical training programmes are based on provoking a state of transient fatigue, thus increasing the body’s regenerative capacity and inducing overcompensation of the biological systems involved [1]. Specifically, strenuous exercise, defined as any activity that expends six metabolic equivalents (METS) per minute or more, causes structural damage to muscle cells, leading to pain and swelling, increased free radicals, impaired immune function and the removal of proteins from circulation, among other consequences [2,3,4]. These processes have various clinical manifestations, including inflammation and immunosuppression, which heightens vulnerability to infection [1].

Therefore, if the imbalance between the work and recovery phases is prolonged, the organism may be unable to adapt properly to the physical workload. The resulting state of excess training has negative consequences not only for physical performance but also for health [1] and when this state occurs, reactive oxygen species (ROS) are synthesised in the body [5,6]. Many studies have focused on ROS, but it is only now that the essential role they play in cell equilibrium and homeostasis is beginning to be understood [6,7], they also intervene in muscle contraction, provoking adaptive responses in muscle fibres [6]. Recent studies, such as Torre et al. [8] and Fang and Nasir [9], have shown that ROS induced by physical exercise can regulate enzymatic and non-enzymatic antioxidants in the biological system. Therefore, exercise could be an optimiser of ROS, preventing cellular oxidative damage, and these molecules also provide a signal for muscle adaptation [10,11]. Moreover, they act as molecular messengers, interacting with proteins that are sensitive to the oxidation–reduction reactions (REDOX) that regulate different processes in the body, such as insulin sensitivity, growth factor signalling, vasodilation and immune response [8].

Exercise may also produce alterations in the immune system, as neutrophils enter the damaged tissue in order to eliminate necrotic tissue [7]. This system intervenes in stress response through pro-inflammatory mediators produced by macrophages resident in tissues, especially the muscles, and T lymphocytes. Some of these mediators, such as interleukin 6 (IL-6), exert local actions in damaged tissues when they are produced in high amounts, as is the case in intense exercise [1]. They can also amplify the signal at the systemic level, generating the acute phase response to tissue damage and participating in the stimulation of the neuro-endocrine system through pathways parallel to the hypothalamic–pituitary–adrenal axis, in reactions that are sometimes disproportionate and harmful [1,12]. An increase in the synthesis of the hormones creatinkinase (CK), glutamic oxaloacetic transaminase (GOT) and glutamic-pyruvic transaminase (GPT) indicate recent muscle damage and the addition of external antioxidant substances can reduce their production [13].

Aware of the critical role played by inflammation in eccentric exercise-induced muscle damage, researchers have long focused on containing and reducing inflammatory responses by pharmaceutical, physical and nutritional means [14]. In this respect, dietary antioxidants are useful to stabilise free radicals and the ROS produced by oxidative stress during physical exercise [8]. They also contribute to optimising the adaptive response of biological systems to physical exercise [2]. An immunomodulator can be defined as a substance that alters the immune response, either increasing or decreasing the ability of the immune system to produce specific serum antibodies or sensitised cells that recognise and react with the antigens that initiate their production. Some of these substances are natural and others are of pharmacological origin [1].

One such dietary antioxidant is vitamin C, which interacts directly with free radicals and reduces the lipid peroxidation that can damage the cell membrane [8,15]. Lipid peroxidation may also be decreased by vitamin E. Both of these vitamins act as direct antioxidants, regulating REDOX levels through the elimination of ROS [8]. However, a wide variety of immunomodulatory substances with antioxidant capacities are currently used as oral supplementation, some of which are analysed in this review.

Coenzyme Q_10_ (Co Q), also know as ubiquinone, is a fatsoluble benzoquinone found in most eukaryotic cells. This molecule participates in the electron transport chain producing ATP, hence its powerful antioxidant and cellular protective role [16]. Melatonin, a hormone synthesized by the pineal gland, has antioxidant properties that include scanvenging for free radicals and regulating the activity of antioxidant enzymes [17]. Another substance that could counteract the negative effects of physical exercise is oats (*Avena sativa*) as they may have a protective effect against cell apoptosis caused by oxidative stress [18,19,20].

In view of these considerations, the aim of the present study is to determine recent advances regarding antioxidant effects of different oral substances after performing muscle strength exercises that induce an increase in markers of oxidative stress and muscle damage (IL-6, CK, GOT) [1,13].

## 2. Materials and Methods

The Preferred Reporting Items for Systematic Review and Meta-Analyses guidelines (PRISMA) were used. The databases used were Web of Science, Medline, PubMed and SportDiscus.

### 2.1. Study Search

Two authors (C.C.-G. and I.M.G.-R) conducted the search. The databases used were Web of Science, MedLine, PubMed and SportDiscus in English and Spanish. The search dates were limited to the period 1 January 2011 to 1 January 2021. The search strategy followed was: (“Antioxidant” OR “Antioxidant Effects” OR “Antioxidant Supplements” OR “Nutrition Supplementation” OR “Anti-Oxidants” [All Fields]) AND (“Resistance Training” OR “Strength Training” OR “Eccentric Exercise” OR “ Eccentric Training” OR “Concentric Exercise” OR “Concentric Training” OR “Isometric Training” OR “Isometric Exercise” OR “Muscle Strength” OR “Isokinetic Training” OR “Isokinetic Exercise” OR “Weight Bearing Exercise” [All Fields]) AND (“Inflammation” OR “Muscle Weakness” OR “Muscle Damage” OR “Muscle Fatigue” OR “Oxidative Stress” OR “Oxidative Stresses” [All Fields]).

### 2.2. Eligibility Criteria

Articles that met the following criteria were included in this review: (I) randomised control study (RCT) design, (II) dietary supplementation included oral antioxidant substances, (III) only studies that used resistance training and aerobic exercises as a form of intervention were included. Studies that met the following criteria were excluded: (I) published in a language other than English or Spanish; (II) full access to the text and conference presentations, theses and books were not contained; they were duplicates, editorials, review papers or expert opinions; (III) the primary and/or secondary authors of the articles did not respond to email requests to provide missing and required information.

### 2.3. Study Selection

After an initial screening for our inclusion and exclusion criteria and the removal of duplicates (C.C.-G. and I.M.G.-R.), three reviewers independently assessed the articles by screening abstracts (C.C.-G., D.J.-M. and I.M.G.-R.). Next, the full text of each article was obtained and screened against the exclusion criteria. Any disagreement between them was resolved through discussion or the intervention of an arbitrary third investigator was requested. After this first stage, the previously selected articles were read in full. Subsequently, the selected articles were evaluated by two researchers (C.C.-G. and I.M.G.-R.) to review the evaluation of methodological quality using the PEDro scale.

### 2.4. Data Extraction

An Excel form was used for data extraction. Of each manuscript selected for review, the following information was considered: study, sample, participants, training methodology, supplementation, method, administered doses, measurements, results.

### 2.5. Assessment of Methodological Quality

The quality of the evidence of the articles included in this review was assessed using the PEDro scale, which is based on criteria that identify whether RCTs have sufficient internal validity and statistical information to interpret the results (external validity (Item 1), internal validity (Items 2–9) and statistical reporting (Items 10–11). Each item is classified as yes or no (1 or 0) according to whether the criterion is clearly met in the study. The total score is from Item 2 to 11, so the maximum score is 10. Two independent investigators (C.C.-G. and I.M.G.-R.) evaluated the articles using this scale. In case of discrepancy, a third evaluator (D.J.-M.) was consulted.

### 2.6. Stadistical Analysis

The agreement rate between the reviewers for the quality assesment of the studies was calculated using kappa statistics. Agreement was assessed as proposed by Landis and Koch (1977), with 0.8 to 1.0 indicating almost perfect agreement, 0.6 to 0.8 substantial, 0.4 to 0.6 moderate, 0.2 to 0.4 fair, zero to 0.2 slight and zero or lower poor. The software package used was SPSS statistics (IBM Corporation, Armonk, NY, USA) and the result was 95% agreement—almost perfect (Table 1).

## 3. Results

### 3.1. Article Selection

The database search identified a total of 1709 articles. After duplicate studies were eliminated, 1690 remained. In each case, the title and abstract were evaluated according to their relevance to the review criteria. Studies that were duplicates, those for which the full text was not available, those carried out with animals, those not related to the search and those based on subjects who had a pathology or injury at the time of the study were all eliminated. After applying these criteria, 31 articles remained. However, in eight there was no control group, in another ten no antioxidant was consumed, and in another seven the state of fatigue did not result from physical exertion. Finally, therefore, seven articles were selected for inclusion in this systematic review (Figure 1).

### 3.2. Included Studies and Study Characteristics

Table 2 presents the main characteristics of the included studies. Finally, seven articles met the inclusion criteria established for this review, of which only one included men and women in the study. Of the other studies, four were conducted only with men, one only with women, and another study did not specify the sex of the participants in the study sample. The age range of the participants was from 20 ± 0.7 years to 39 ± 8.7 years; the weight from 62 ± 8 kg to 80 ± 10 kg; the height from 167 ± 7 cm to 179 ± 6.04 cm. In the study by Thang et al. [14] weight and height were not specified; BMI and maximum heart rate results were shown. In the study by Ortiz-Franco et al. [17] and Sarmiento et al. [16] the age, weight and height are indicated both in the experimental group and in the control group.

Table 3 presents the summary of the selected studies investigating antioxidant supplementation after strength exercise. This includes participants, training methodology, strength exercises, supplementation, method, administered doses, measurements, biochemical data and results.

### 3.3. Methodological Quality and Risk of Bias

Table 4 shows that of the seven articles selected for analysis, the three that scored most highly on the PEDro scale (8/10) were Ortiz-Franco et al. [17], Sarmiento et al. [16] and Zhang et al. [14]. The following criteria were met by all seven studies: the selection criteria were specified (Item 1); the allocation to the different groups was hidden (Item 3); as concerns the most important prognostic indicators, the groups were similar at baseline (Item 4); and all subjects were blinded (Item 5). With respect to Item 2 (the subjects were randomly assigned to the groups), only Ammar et al. [21] failed to meet this requirement. Item 6 (all those who administered the therapy were blinded) was only met by Da Silva et al. [22] and Ortiz-Franco et al. [17]. Item 7 (all the assessors who measured at least one key outcome were blinded) was only met by Da Silva et al. [22] and Zhang et al. [14]. Item 8 (the measures of at least one of the key results were obtained from more than 85% of the subjects initially assigned to the groups) was met by every paper except Leonardo-Mendonςa et al. [23]. Item 9 (the results of all subjects who received treatment were published) was met by all studies except Ammar et al. [21] and Da Silva et al. [22]. Item 10 (between-group statistical comparisons were made and at least one key result was reported) was not met by Da Silva et al. [22], Ortiz-Franco et al. [17] or Zhang et al. [14]. Finally, Item 11 (the study included specific measures of variability for at least one key result) was met by all papers except Da Silva et al. [22] and Zhang et al. [14].

### 3.4. Presentation of Results

Regarding the sample size, Ammar et al. [21] are those with the smallest number, nine participants, and Sarmiento et al. [16] conducted the study with 100 subjects (the largest sample size). The other studies included a total of 10, 14, 21 and 24 subjetcs [14,17,23,24]. The mean age of the participants in the selected studies ranged from 20 years to 38.9 years. The physical fitness level of the participants was low in the studies by Da Silva et al. [22] and Thang et al. [14]—they only required their subjects be healthy. It is also in this last study that the data on the weight and height of the participants were not collected.

The studies by Mcleay et al. [24] and Ortiz-Franco et al. [17] also requested a physical examination and a medical interview. Regarding the training methodology, in each study a different kind of training was carried out for a certain time. In terms of the antioxidant supplementation used, two of the studies used melatonin [17,23] and the other five all used different substances. Four blood samples were drawn in all studies except two [14,16], which made five and six measurements respectively. Two of the studies used weightlifting as the exercise [21,22], while another, in addition to this, performed an hour of endurance and aerobic running [23]. In studies that used weightlifting as the exercise, the first indicates repetitions and percentage, 2 sets of 3 repetitions at 85% and 3 sets of 2 repetitions at 90% [21]. Two other investigations used strength exercises and quadricep contractions as the sports training [16,24]. In the latter two, in which blueberries and Coenzyme Q_10_ were used as antioxidant supplements, respectively, there was an adaptation of the antioxidant processes and a regulation of oxidative stress. The study by Thang et al. [14] is the only one that used downhill running as training and they obtained an improvement in the inflammatory response thanks to supplementation with oatmeal. This grain can be used like an antioxidant supplement because one of the major components of polyphenolic amides (nonflavonoids) is AVA, which has beneficial effect on the organism after training [14]. The study carried out by Da Silva et al. [22], who used taurine as the supplement, found that although it reduces muscle damage it does not decrease the inflammatory response.

In the study carried out by Leonardo-Mendonςa et al. [23] with oral melatonin supplementation, skeletal muscle protection occurred after endurance training, weights and aerobic running. Iin the study by Thang et al. [14] a five-minute pre-training warm-up was mentioned, followed by four sets of fifteen minutes running downhill with a gradient of −10%. Furthermore, in the study by Ammar et al. [21] the first two sets of three repetitions at 85% and the following three sets of two repetitions at 90% involved weightlifting. This led to a better recovery of acute and delayed responses to oxidative stress.

The abovementioned antioxidant power substances with a training programme that promotes muscle damage led to the inclusion of these studies in this systematic review.

#### 3.4.1. Dietary Intake

In the studies by Da Silva et al. and Leonardo-Mendoça et al. [22,23], the dietary intake of the participants was recorded for at least three consecutive days. In the first mentioned study, food intake was also recorded on the day of the test and one day after. In the study by Ortiz-Franco et al. [17], a nutritional assessment was performed by recording dietary intake for fourteen days.

#### 3.4.2. Blood Samples

Two studies analysed enzymatic antioxidants (CAT, catalase and GPx, glutation peroxidase) [21,22] and the latter also analysed the antioxidant enzyme SOD, superoxide dismutase. These are not altered by taurine supplementation neither are the inflammatory enzymes (TNF-α, tumor necrosis factor-α, IL-1β, interleukin-1 β and IL-10, interleukin-10). In another study, in addition to the general analysis of markers of muscle damage and oxidative stress, lymphocytes were isolated [17]. In the study by Sarmiento et al. [16], the amount of lactate analysed increased in both groups (control and experimental) even when ubiquinol was ingested. The study by Thang et al. [14] was the only one to measure cell adhesion molecule (sVCAM-1), colony stimulating factor (G-CSF) and chemotactic cytokine (MCP-1), which used oatmeal as an antioxidant supplement.

## 4. Discussion

The aim of this study is to determine the effect of antioxidant consumption on markers of oxidative stress and muscle damage after performing muscle strength exercises. At the general level, supplementation with pomegranate juice alleviates oxidative stress, taurine reduces muscle damage, melatonin protects the skeletal muscles, blueberries decrease oxidation and oats mitigate muscle damage. In relation to high-intensity training, the results obtained cannot be grouped due to the wide diversity of the studies addressed and we did not find other studies that followed the same training protocol while incorporating different antioxidant substances. In this respect, different studies used weights as eccentric exercises [23,25,26,27]; however, in each of them a different training protocol was performed; increasing weight percentages in each repetition or adding another exercise to the programme [28,29]. Concerning design, all studies were randomized and double-blind. Selected subjects had to be physically active except for two studies that only required them to be healthy [14,29,30,31].

Our analysis of the studies selected highlights the benefits of oral antioxidants, which provide varying levels of improvement depending on the antioxidant used.

Substances currently used as oral antioxidant supplements could produce improvements in the body by decreasing the markers of oxidative stress or muscle damage by restoring the REDOX balance or by protecting the skeletal muscles [32,33,34,35].

Regular physical exercise is associated with health benefits and decreased mortality [16,36,37,38]. However, strenuous exercise generates oxidative stress and a pro-inflammatory state, among other negative consequences [1,38,39,40]. In view of the major impact of inflammation on the organism, supplementation with immunomodulatory substances would be beneficial [41,42,43,44,45]. Among many populations there is growing interest in regular physical exercise, often involving muscle strength training exercises [46,47,48,49,50,51]. This pattern of activity highlights the importance of incorporating antioxidant substances in the diet to counteract the possible negative effects of this exercise on the body [17,50].

Among other consequences, muscular strength exercises may provoke alterations in the REDOX reactions within the immune system via the infiltration of neutrophils into the damaged tissues, leading to inflammation in the muscles and increased markers of oxidative stress [7,8,14]. The studies reviewed [14,16,17,21,22,23,24] concerned different antioxidants examined and their effectiveness at counteracting the negative effects of muscle strength training. In the seven studies reviewed, this hypothesis was tested by associating each oral antioxidant with a specific action at the systemic level. In every case, an improvement was achieved and reflected as a decrease in the markers of muscle damage and/or oxidative stress.

In Ammar et al. [21], after pomegranate juice was used as a supplement (three tablets a day during the 48 h prior to each training session) improvements were recorded in the recovery from acute and delayed oxidative stress. Specifically, there was a reduction in the immediate increase in malondialdehyde (MDA), which produced a significant effect on lipid peroxidation and antioxidant parameters.

These findings corroborate Mazani et al. [49], who showed that 240 mL of pomegranate juice consumed daily for 14 days before using a treadmill significantly increased the activity of enzymatic antioxidants. Therefore the antioxidant properties of pomegranate juice are linked to the attenuation of lipid peroxidation responses since they lead to a reduction in acute oxidative stress. In view of these findings, it is suggested that antioxidant activities have the greatest effect immediately after intensive physical exercise, when the pomegranate juice supplementation was provided [21]. In the latter study, unlike Ammar et al. [21], the pomegranate juice was supplied in the form of tablets ingested two days before the session. Da Silva et al. [22] showed that supplementation with taurine, one tablet a day for 21 days, reduced oxidative muscle damage but not the inflammatory response. In this research, the CK parameter (a marker of skeletal muscle injury) was used to measure muscle damage after extensive exercise. The authors concluded that taurine affects vascular tone, producing vasodilation and improving blood flow and thereby reducing ischaemia and the production of ROS [22].

On the other hand, studies in which melatonin supplementation was examined observed that it helped to prevent extracellular and intracellular oxidation and protected the skeletal muscles against oxidative damage, as well as reducing the damage from high intensity training [17,23]. Regarding antioxidant adaptation processes, it was observed that blueberries helped regulate antioxidant adaptation processes [24,50]. Melatonin protects against muscle damage and oxidation caused by free raedicals. One of the most attractive properties of this hormone that distinguishes it from other antioxidants is that it has the ability to scavenge ROS (reactive oxygen species) and reactive nitrogen species [23]. The blueberry fruit used by Mcleay et al. has a high free radical scavenging capacity, so it would not only reduce oxidative stress but also inflammation [24].

During training, oxidation and oxidative stress could be reduced by daily coenzyme Q_10_ intake [16].

Additionally, studies on oat intake [14,51] have concluded that this substance improved the inflammatory response of plasma and mitigated muscle damage (decreasing the CK marker in plasma), thus reducing circulatory inflammatory cytosines and inhibiting chemokine expression. Specifically, the study by Zhang et al. [14] was the only one of those reviewed that recorded the pain sensations experienced (probably related to the muscle damage induced by high intensity training). This sensation was reduced when the diet contained oats, which suggests that long-term oatmeal consumption provides specific protection against muscle damage and oxidative stress.

The present study has certain limitations, notably the small number of papers selected for analysis, which prevented us from examining the parameters considered in greater detail. This problem reflects the fact that, to date little research has been conducted with respect to muscle strength training [22,23,24,25,26]. In this respect, we believe that it would be useful to make a comparable comparison between studies if the same training protocols were used, performing the same eccentric exercises but with the same repetitions and the same intensity, differing only in the antioxidant supplements used. In this way, it would be easier to appreciate the positive effect of different oral antioxidant substances and the practice of physical exercise. Furthermore, most of the studies included in our review were based on a relatively small sample size except for the study by Sarmiento et al. [16], which included a sample of 100 subjects. As mentioned above, the fact that a different training protocol was applied in each study also prevented us from establishing a uniform research pattern. Another possible limitation is the heterogeneous dietary intake of the participants prior to the experiment and the sleep control applied (this consideration is important; according to Finan et al. [48], a lack of sleep can increase pain sensitivity and perception).

Overall, little importance was given to the dietary intake of participants during these studies, but this should be taken into account as diet and consumption of certain substances can influence the markers of oxidative stress and the effect of the antioxidants consumed.

We propose a standardized diet be provided to all participants at the start of the study so that subjects consume the same amount of proteins, fats, vitamins, etc., in a controlled manner so it does not influence the study.

With respect to sleep control, we think that it can influence how pain is perceived, but it is not very cost-effective or accurate. Therefore, it should only be taken into account when studying blood markers related to muscle damage and oxidative stress.

Among the strengths of the study is the fact that the literature search spanned the last 10 years and thus obtained up-to-date results and methodologies. Additionally, the comparison made of the effects of different antioxidants within the same study is valuable.

We invite other researchers to make use of the data presented in this review to conduct future investigations regarding the association between the intake of antioxidant substances and the impact of high intensity physical training. Our recommendations are to include the variable of high intensity training, to increase the size of the study sample, to control the participants’ dietary intake prior to starting the training programme and to monitor the consumption of antioxidant substances.

## 5. Conclusions

From the results obtained, we observe that supplementation with antioxidant substances could help regulate cellular homeostasis, thus contributing to better REDOX in the skeletal muscles.

In view of these considerations, we conclude that there exists a positive relationship between the consumption of supplements with antioxidant properties and reduced cellular damage and inflammation.

Comparisons of our findings with other meta-analyses and systematic reviews addressing the question of antioxidant supplementation for persons subjected to a strenuous exercise protocol shows that there is a positive relationship between this supplementation and post-exercise recovery. In other words, the antioxidant substances ingested during a given period play a transcendental role in the subject’s post-training recovery [6,9,10,16]. At a practical level, we also observe that the antioxidant substances examined in our research can be found in the form of tablets, which makes it easier to control the exact dose being ingested [16,17,22,23]. Moreover, this presentation makes the supplements readily accessible to the general population, as well as those engaged in strenuous exercise.

The findings of this systematic review indicate that there is a very low number of studies looking at muscle damage and oxidative stress while performing muscle strength exercises. In summary, the results of this study point to the conclusion that muscle strength exercises produce oxidative stress and that appropriate supplementation with antioxidant substances may prove to have beneficial effects, thus improving the antioxidant status and preventing muscle damage induced by strenuous exercise. Future research should assess these same variables with a larger number of participants while taking into account the variable of muscular strength exercises.

## Figures and Tables

**Figure 1 ijerph-19-01803-f001:**
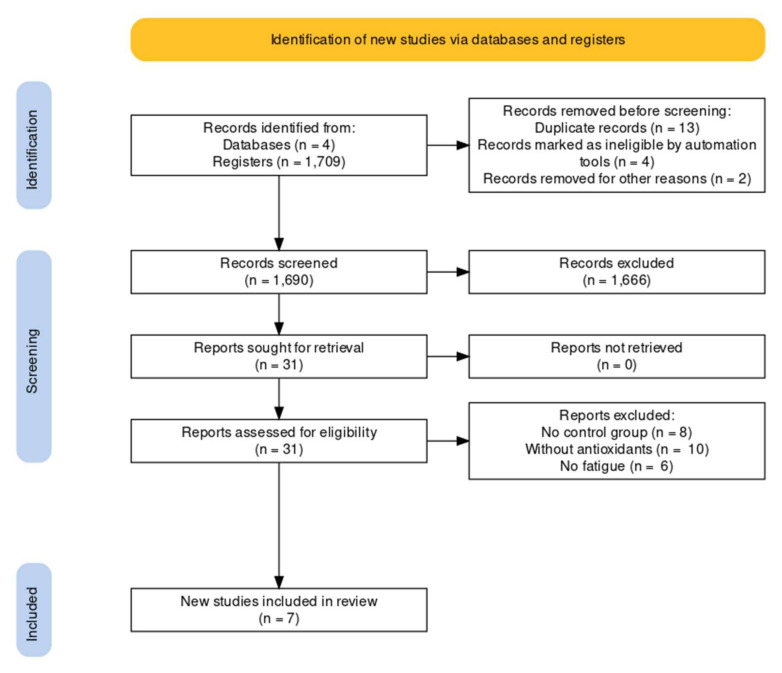
PRISMA flow diagram of the selection process of articles included in the systematic literature review of investigating the effect of antioxidant supplementation after a strength exercise.

**Table 1 ijerph-19-01803-t001:** Interpretation of Kappa Statistic.

Kappa Statistic	Strength of Agreement
<0.00	Poor
0.00–0.20	Slight
0.21–0.40	Fair
0.41–0.60	Moderate
0.61–0.80	Substantial
0.81–1.00	Almost Perfect

Agreement measures for categorical data.

**Table 2 ijerph-19-01803-t002:** Characteristics of the participants of the studies selected for the review.

Author (N)	Sample Size and Sex (Male/Female)	Age (Mean ± SD)	Weight (kg)	Height (cm)	Level of Condition of Physical Activity or Health
Ammar et al., 2017 (9)	9/0	21 ± 1	80 ± 10	175 ± 0.08	Physically active
Da Silva et al., 2014 (21)	21/0	21 ± 6	78.2 ± 5	176 ± 7	Healthy
Leonardo-Mendoça et al., 2017 (24)	24/0	20.3 ± 0.71	74.7 ± 3.22	176 ± 1.83	Physically active
McLeay et al., 2012 (10)	0/10	22 ± 1	62 ± 8	167 ± 5	Physically active
Ortiz-Franco et al., 2017 (14)	14/0	Placebo Group 28.43 ± 4.39	78.39 ± 6.68	176 ± 3.98	Physically active
		Melatonin Group 26 ± 6.03	79.96 ± 7.29	179 ± 6.04	
Sarmiento et al., 2016 (100)	100/NA	Placebo Group 38.2 ± 7.7	74.8 ± 9.8	174 ± 7.6	Physically active
		Ubiquinol Group 38.9 ± 8.7	76.8 ± 8.9	175 ± 5.0	
Thang et al., 2020 (24)	11/13	23 ± 1.2	NA	NA	Healthy

NA = not available.

**Table 3 ijerph-19-01803-t003:** Summary of studies investigating antioxidant supplementation after strength exercise.

Study	Participants	Training Methodology	Strength Exercises	Supplementation	Method	Administered Doses	Measurements	Biochemical Data	Results
Ammar et al., 2017	Trained at least 5 sessions per week, 3 years of weightlifting experience, no injuries, no anti-inflammatory	2 sets of 3 reps at 85% weightlifting and 3 sets of 2 reps at 90%	3 Olympic Weightlifting exercises	Pomegranate juice	3 tablets per day (48 h before each of the sessions)	250 mL or 3 tablets	At rest and 3 minutes and 48 hours after each session	MDA, CAT, GPX, UA, Tbil	Improved recovery of acute and delayed responses to oxidative stress
Da Silva et al., 2014	No smoking, no antioxidants or taurine, no resistance training for at least 6 months, no injury or illness	Eccentric exercise, weight lifting for 14 days	The subject´s one repetition maximum by elbow flexors and extensors	Taurine	Once daily for 21 days	50 mg per kg mass per day for 21 days	Days 16, 18 and 21 during training	Xylenol orange, protein carbonylation, total thiol content, superoxide dismutase, CAT, GPX, TNF-α, IL-1 β, IL-10	Improves performance, reduces muscle damage and oxidative stress but does not decrease inflammatory response
Leonardo-Mendoça et al., 2017	Healthy, non-smokers, no medication or supplementation	8 one-hour sessions per week (resistance, weights and aerobic running). Total of 10 h per week	2 sessions weight training	Melatonin	For 4 weeks, 30–60 min before bedtime	100 mg per day	Before starting the study and at the end of the supplementation	Glucose, total cholesterol, HDL and LDL cholesterol, triglycerides, urea, creatinine, uric acid, AST, ALT, CK, LDH	Prevents extracellular and intracellular oxidation, protection of skeletal muscle against oxidative damage
McLeay et al., 2012	Physically active, resistance and aerobic exercise twice weekly, at least 1 year’s experience, health questionnaire	300 eccentric, isometric and concentric quadricep contractions	300 contractions of the quadriceps	Blueberries	Morning, noon and afternoon	Each smoothie blended 200 g blueberries (total: 1 kg of blueberries)	12, 36 and 60 h after exercise	CK, plasma protin carbonyls, plasma radical oxygen species, IL-6, plasma antioxidant capacity	Accelerates the recovery of maximum muscle isometric strength and regulation of antioxidant adaptation processes
Ortiz-Franco et al., 2017	Medical interview, non-smoker, no lactose intolerance, no medication, regular sleep schedule	6 sessions per week of 60–75 min per day (HIIT and strength exercises)	3 sets of 10 repetitions at 70–80% of 1RM	Melatonin	1 daily dose before exercise	20 mg daily	Before the start of the study, immediately after and 24 h after the physical exercise	Glucose, urea, creatine, uric acid, total cholesterol, HDL, LDL, Triglycerides, total bilirubin, iron, albumin, prealbumin, transferrin, ferritin, red blood cells, haemoglobin, haematocrit	Improves antioxidant status and beneficial effects on damage produced by high intensity training
Sarmiento et al., 2016	Firefighters, medical interview and physical exam	Circuit of 10 bodybuilding exercises (sports press, chest press, seated row, shoulder press, hamstring curl, chest press, chest step, chest surveyor, push with weight and quadriceps extension	Chest press, shoulders press, femoral biceps flexion, quadriceps extension	Coenzyme Q_10_	For 2 weeks prior to the exercise protocol	200 mg daily	5 samples in total (before supplementation, after supplementation, after exercise, after 24 h of rest and after the second exercise test	8OHdG, lipid peroxides, LDL oxidized, carbonyl	Decreases oxidation and does not increase oxidative stress
Thang et al., 2020	Non-obese, no gastrointestinal problems or pathologies, non-consumer of tobacco or alcohol, not allergic to oatmeal products or AINEs	5-minute warmup, then 4 series of 15 min downhill running at a gradient of −10%, intensity equivalent to 75% of max.HR.	Treadmill at 75% of max.HR.	Oatmeal	12 units daily for 8 weeks	30 g of oatmeal	6 samples total (at rest, post test, after 4, 24, 48 and 72 h)	IL-6, IL-1RA, sVCAM-1 cell adhesion molecule, G-CSF, MCP-1, CK	Improved plasma inflammatory response to exercise stress and mitigated muscle damage

Abbreviated biochemical data: MDA (malonaldehyde), CAT (catalase), GPX (glutation peroxidase), UA (uric acid), Tbil (total bilirubin), TNF-α (tumor necrosis factor-α), IL-1 β (interleukin 1-β), IL-10 (interleukin-10), AST (low-density lipoproteins), ALT (alainne aminotransferase), CK (creatin kinase), LDH (lactate dehydrogenase), IL-6 (interleukin-6), IL-1RA (anti-inflammatory cytokine), GCSF (col-ony stimulating factor), MCP-1 (chemotactic cytokine).

**Table 4 ijerph-19-01803-t004:** Evaluation of the quality of evidence PEDro scale.

Authors	1	2	3	4	5	6	7	8	9	10	11	Total
Ammar A et al.	1	0	1	1	1	0	0	1	0	1	1	6/10
Da Silva LA et al.	1	1	1	1	1	1	1	1	0	0	0	7/10
Leonardo-Mendonςa R et al.	1	1	1	1	1	0	0	0	1	1	1	7/10
McLeay Y et al.	1	1	1	1	1	0	0	1	1	1	1	8/10
Ortiz-Franco M et al.	1	1	1	1	1	1	0	1	1	0	1	8/10
Sarmiento A et al.	1	1	1	1	1	0	0	1	1	1	1	8/10
Zhang T et al.	1	1	1	1	1	0	1	1	1	0	0	7/10

PEDro items: (1) Eligibility criteria; (2) Random allocation; (3) Concealed allocation; (4) Comparability at baseline; (5) Patient blinding; (6) Therapist blinding; (7) Assessor blinding; (8) At least 85% follow-up; (9) Intention to treat analysis; (10) Between-group statistical comparisons; (11) Point measures and measures of variability.

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
