# Peer review of "Effect of Antioxidant Supplementation on Markers of Oxidative Stress and Muscle Damage after Strength Exercise: A Systematic Review"

_ijerph, 2022, doi:10.3390/ijerph19031803_

Round 1

Reviewer 1 Report

Congratulations to the authors. This is a practical review and the results are very interesting. The objective of the paper was to determine the effect of antioxidant consumption on markers of oxidative stress and muscle damage after performing muscle strength exercise. The paper is very well written, data analysed and presented in an appropriate and clear manner. I have highlighted some specific comments which could be considered and may improve the paper in each section.

Introduction

- It would be pertinent to speak of biochemical markers of muscle damage such as CK, GOT and GPT. In this sense, it would be interesting to know how antioxidants can regulate these parameters.

- The second and third paragraphs could be joined to form a larger and more meaningful paragraph within the text.

- Line 94: Please use in addition or however, but not both together.

Materials and Methods

- Please enter the databases in which the literature search was carried out.

Results

- The paragraph referring to point number 3.1 seems to have a formatting error.

- Table 1: The column "author" and "age" could be expanded to better read the information.

- Table 2: It is a very complete and clarifying table, however the "Biochemical data" column would need a space between study and study to be able to understand clearly.

  • I would give a single section within the results to "strength programs or strength exercises" since it is the most relevant aspect on which the review is based.

Discussion

- The discussion is well structured, establishing the most determining physiological mechanisms caused by the intake of antioxidants in muscle damage.

 Limitations and Conclusion

- The limitations are well defined and are important when interpreting the results.

- Line 373-374: This phrase should be deleted as it says the same as in the first paragraph of the conclusion.

Author Response

Dear Reviewer - I am attaching a document with the response to your comments. Thank you for your time and assistance.

Reviewer 2 Report

Overall I congratulate the authors on attempting to address the effects of antioxidant supplementation on muscle damage via a systematic review, however whilst the authors have attempted to address this research question, there are some areas that I feel the manuscript needs to address in more detail. 

I have provided detailed comments below on where I feel the authors may wish to address these general comments in the manuscript. 

Abstract: 

Line 19: Opening word (the) needs to have a capital letter (i.e. The purpose....) for consistency within other sections. 

Line 19: I would remove the second 'the' from the sentence so it reads: '...was to determine effects of antioxidant....'

Line 23-24: A sentence regarding removal of duplications and the application of the exclusion criteria to get to the n=7 papers is needed here. 

Introduction: 

Generally well-written, however some issues regarding specificity of information to streamline the introduction to match the purpose of the review is needed. I have provided more comments below:

Line 33-40: I understand what the authors are attempting here, however this paragraph has very limited link to the overall objective of the review - it provides more of a 'general' overview of exercise and benefits rather than being specific to muscle damage etc. which would focus the introduction of the review from the start. I feel this section could easily be removed and would save on word count to be used in other areas of the manuscript. 

Line 42-60: A number of the paragraphs can be merged in to one paragraph. In the current format, of the short/sharp sentences, hampers the readability of this section. 

Line 73-78: This information needs to be introduced within the introduction earlier as this is important information relating to the purpose of the review. 

Line 91-97: This section lacks some detail with regards to the included antioxidants within the review. The authors mention Vit C/E here - and whilst these are well-documented for their antioxidant capacities - neither of these are specifically looked at within the review. Therefore, I feel the authors need to expanding this section to include some further information to incorporate differing examples of antioxidants here to provide further context. 

Line 94-97: Similar to above point, this statement lacks any clarity with regards to the included antioxidants. Addressing the previous point will help with this here. 

Line 98-100: I feel the aim presented here lacks detail. The authors have an opportunity to be specific with regards to specific antioxidants and their effects rather than just 'antioxidant consumption'.

Methods: 

Line 106-115: The authors need to correctly display how the search terms (including boolean operators) were used - if I were to attempt to re-conduct this search, I (should) be able to 'cut & paste' the terms in to a relevant database. In the current format, I am unable to do this. Please amend. 

Example: (“antioxidant” OR “anti-oxidant” [All Fields])

Line 141: Remove the secondary 'full stop' from section title

Line 145-150: Given the fact that multiple authors scored the included studies against the PEDro scale, I am surprised to see that no statistical analysis was carried out to test the inter-rater reliability of the scoring. I would strongly recommend the authors consider the use of a Cohen's Kappa to assess this and report the results accordingly. I would also look to add an additional section within the methods section to reflect this inclusion (i.e. 2.6 Statistical Analysis) and subsequent reporting in the results section. 

Results: 

For me, this is where the manuscript needs an overhaul and for the authors to reconsider the details provided and the structure of this section.

Firstly, as per previous comments, revisions should include the relevant statistical analysis of the inter-rater reliability. 

Secondly, given the inclusion criteria of strength training, I was surprised to see three of the included studies included additional or non-strength training modalities (i.e. goes against inclusion criteria).

Leonardo-Mendonca et al, 2017 = running + weights protocols

Ortiz-Franco et al, 2017 = HIIT + weights

Thang et al, 2020 = running protocol only 

These modalities are not exclusively strength training-related and goes against the inclusion criteria laid out in the methods section. Given the additional modalities, this then further questions the findings of these studies given that it could be argued that with the additional modalities adopted within the study a larger magnitude of muscle damage would be seen vs strength-training exclusively. 

I have further concerns over the use of the Thang et al, 2020 study given that oatmeal was used was the intervention. Whilst it may contain antioxidants, it is a rich source of carbohydrate also, which is also a known mediator of glycogen/muscle recovery, therefore I am not certain that with this, and the exercise protocol adopted within the study, the authors should include this as part of the review. 

Line 156-165: There appears to be a formatting issue with the paragraph, please amend. 

Within Table 2, can the authors please clarify also how many doses of blueberries were given to participants in the McLeay et al, 2012 study? Method states morning and afternoon, however not specifically when. I feel this is important to distinguish given that 1kg (?!?!) of Blueberries was administered throughout. 

The information regarding sample size is duplicated in Table 1 & 2 and is not needed - I would consider removing it from one of these tables. 

In Table 2, I would provide a key underneath so that all abbreviations are listed here for reference as opposed to listed in full then abbreviated within the same column - this will aid readability of the table. 

Line 173 - 177: With the general data presented, for variables such as age, please include the term years (i.e. 20 - 39 years). Additionally, I would round up the decimal places (i.e. 38.9 to 39 years). 

Within this section also, I would prefer to see the mean+SD data of these variables rather than just a range (i.e. Age: XX+XX years, Body Mass: XX+XX kg etc.) 

Table 3: Given the suggested revised approach to the inter-rater reliability, I would look to present all the scores from the authors who scored against the PEDro scale. 

Line 215-216: I would prefer to see a sample size reported for ALL the participants from all the studies here (i.e. the total number of participants included was XXX) as opposed to just mentioning the smallest/largest sample size - this does not add anything in the current format. 

Line 215-243: This links back to previous comments in the introduction/aims of the review - there is a lack of rationale throughout as to why these antioxidants have been included, this needs addressing within the review in the relevant sections (i.e. Intro/Methods) which will help provide context to these findings. 

Additionally, as previously mentioned - the focus is on strength training, however studies have been included that have additional exercise modalities and so the information included in this section (3.4) may need further consideration at Line 231-240. 

Line 250-259: There are a number of abbreviations used in this section that have not been named first, followed by the abbreviation, please amend. 

Discussion: 

Given previous comments regarding including rationales for specific antioxidants, I think the discussion will need to be amended throughout. I have provided some general comments below to assist with this.

Line 265-266: Given previous comments, I think this sentence may need to be addressed regarding discrepancies in study design/protocols etc. 

Line 272-288:  I feel this section of the discussion would further benefit from more balance in relating to the benefits of adaptive responses etc. that may be blunted by antioxidant supplementation., particularly in certain phases of a training cycle (i.e. pre-season period where adaptation is beneficial)

Line 291 & 304: As per previous comments, some abbreviations used without full name provided first (MDA, CK etc.) please amend.

Line 308-329: As previously highlighted, given the suggestions with regards to some of the included studies, this section may need to be revised and aim to synthesise the information as opposed to providing an analysis of each individual study. 

Line 330-339: I feel these limitations need further expanding on to provide more context. For example, can the authors suggest how a standardized protocol and/or dietary intake may be beneficial? In the current format - it lacks some critical analysis. Additionally, throughout the review, sleep has not been discussed or mentioned, however you have it as a limitation? How does sleep link to antioxidant and reducing muscle damage? This needs further discussion

Conclusion: 

Line 356-360: You mention 'compared to other meta-analyses and systematic reviews....' however you have not compared your findings/mentioned these in any great detail within your discussion section, therefore having greater emphasis on these findings vs your own will provide greater depth to your discussion and further support this statement here. 

Author Response

(The authors gave the same response as above.)

Round 2

Reviewer 2 Report

I wish to start by thanking the authorship team for revising their manuscript based upon previous comments. Generally, these have been conducted to a high standard, and I commend the authors for doing this in a timely fashion. Following a review, I would recommend that minor revisions are made before the manuscript is suitable to be published within the IJERPH journal. 

I have provided more detail on the relevant sections below. Once addressed I feel the manuscript would be suitable for publication. 

Abstract:

Line 23 - 26: To condense this down, I would consider using a statement (along the lines of): 'Following duplicate removal and application of exclusion criteria, seven articles were reviewed' - the current revision is quite 'wordy'. 

Introduction: 

Line 68: I like the addition of this sentence within the revised manuscript - I feel it offers offer further context relating to muscle damage, however the abbreviated hormones (CK, GOT, GPT) have not been listed in full before being abbreviated - this is the first time they appear in the text. Please amend. 

Line 100: The ... within the brackets are not required - please amend. 

Methods: 

2.5 Statistical Analysis 

Line 155 - 156: I feel the authors need to provide more information here - what software package was used to conduct the Cohen's kappa? What strength agreement threshold was used to determine level of agreement? This information is lacking. 

For context, I would suggest the authors refer to the following research paper to help determine strength agreement thresholds: 

Landis, J. R., & Koch, G. G., 1977. The measurement of observer agreement for categorical data. Biometrics, 33(1), 159 – 174

Results: 

Line 174: The figure title needs to provide more context and link to PRISMA. Can I suggest using a title (along the line of) 'PRISMA flow diagram of the selection process of articles included in the systematic literature review for investigating XXXX....' 

Line 196 (Table 1): I think the formatting needs to be addressed within this table - I would recommend making the margins of the table wider so the subsequent information (primarily the author names) fit appropriately on the page

Line 200 (Table 2): The information included within this table is now much clearer - I think the authors have done a great job here - however can I suggest they change the title of this table as the information is not a 'characteristic' of the study - it is a summary of included studies. Similar to previous comments, can I suggest a title (something similar to) 'Summary of studies investigating XXXX....' 

Table 1 & 2: I still not convinced by the inclusion of the Thang et al study The authors state in their methods (Line 121) 'only studies that used resistance training as a form of intervention were included.' Running on a treadmill (irrespective of speed) is not a strength or resistance exercise, therefore I feel the authors should consider removing this study from their results or amend the exclusion criteria within the methods section to clarify. 

Line 243 - 247: This revision inserted here is not a direct result from the included studies - it is proposing a mechanism (i.e. free radical scavenging). I feel it would be better placed within the discussion section 

Discussion: 

Overall a well-written section of the revised manuscript - the authors have added context to the various parts from the initially submitted manuscript. 

Author Response

We would like to thank you once again for the considerable time and very concrete suggestions that you made for this paper.  We feel the article has been greatly improved by your suggestions and we are deeply grateful. 
